# Clinical Management of Patients with Gastric MALT Lymphoma: A Gastroenterologist’s Point of View

**DOI:** 10.3390/cancers15153811

**Published:** 2023-07-27

**Authors:** Tamara Matysiak-Budnik, Kateryna Priadko, Céline Bossard, Nicolas Chapelle, Agnès Ruskoné-Fourmestraux

**Affiliations:** 1IMAD, Hepato-Gastroenterology & Digestive Oncology, University Hospital of Nantes, 44093 Nantes, France; kateryna.priadko@unicampania.it (K.P.); nicolas.chapelle@chu-nantes.fr (N.C.); 2Inserm, CHU Nantes, University of Nantes, Centre de Recherche en Transplantation et Immunologie, UMR 1064, ITUN, 44000 Nantes, France; 3Hepato-Gastroenterology Unit, University Hospital Universita degli Studi della Campania “Luigi Vanvitelli”, 80138 Naples, Italy; 4Institut of Histopathology (IHP), 44104 Nantes, France; celine.bossardwanadoo.fr@orange.fr; 5Department of Gastroenterology, Saint Antoine Hospital, AP-HP, 75012 Paris, France; aruskonefourmestraux@gmail.com

**Keywords:** MALT lymphoma, *Helicobacter pylori*, “watch and wait” strategy

## Abstract

**Simple Summary:**

Gastric mucosa-associated lymphoid tissue (MALT) lymphomas represent rare gastric neoplasia usually localized and of indolent course, characterized by an abnormal proliferation of small B lymphocytes within the gastric mucosa, and derived from an acquired MALT as a result of chronic inflammation most frequently induced by chronic infection with *Helicobacter pylori*. Their clinical management includes proper staging and eradication of *Helicobacter pylori*, followed by other treatments if necessary, as well as appropriate surveillance, given an increased risk of gastric adenocarcinoma and other cancers in this context. The aim of this review is to describe the latest achievements in the clinical management of these lymphomas with a special emphasis on the role of the Gastroenterologist.

**Abstract:**

Gastric mucosa-associated lymphoid tissue (MALT) lymphomas (GML) are non-Hodgkin lymphomas arising from the marginal zone of the lymphoid tissue of the stomach. They are usually induced by chronic infection with *Helicobacter pylori (H. pylori)*; however, *H. pylori*-negative GML is of increasing incidence. The diagnosis of GML is based on histological examination of gastric biopsies, but the role of upper endoscopy is crucial since it is the first step in the diagnostic process and, with currently available novel endoscopic techniques, may even allow an in vivo diagnosis of GML per se. The treatment of GML, which is usually localized, always includes the eradication of *H. pylori*, which should be performed even in *H. pylori*-negative GML. In the case of GML persistence after eradication of the bacteria, low-dose radiotherapy may be proposed, while systemic treatments (immunochemotherapy) should be reserved for very rare disseminated cases. In GML patients, at diagnosis but even after complete remission, special attention must be paid to an increased risk of gastric adenocarcinoma, especially in the presence of associated gastric precancerous lesions (gastric atrophy and gastric intestinal metaplasia), which requires adequate endoscopic surveillance of these patients.

## 1. Introduction

Gastric mucosa-associated lymphoid tissue (MALT) lymphomas (GML) are non-Hodgkin lymphomas arising from the marginal zone of the lymphoid tissue of the gastric mucosa, usually induced by chronic infection with *Helicobacter pylori* (*H. pylori*) [1,2]. GML are rare, representing less than 1% of all gastric malignant tumors, and they are usually of indolent course. The incidence of GML seemed to increase until the 1990s, and then started to decrease, likely in relation to a decreasing prevalence of *H. pylori* infection [3,4,5]. A new phenomenon is a relative increase in the incidence of *H. pylori*-negative GML, which in some recent studies, accounts for up to 40% of all GML cases [6,7]. The clinical management of GML was not standardized until the publication of several international guidelines in the 2010s [8,9,10,11,12]. The GML treatment strategies have evolved considerably during the last three decades, moving from aggressive (gastrectomy) [13,14,15] to more conservative (*H. pylori* eradication +/− local low-dose radiotherapy or immunochemotherapy) treatments [16,17,18,19,20]. This evolution was possible due to our increasing knowledge of the pathogenesis, natural history, and evolution of these lymphomas, indicating an overall indolent course and low risk of transformation into more aggressive lymphomas, like diffuse large B cell lymphoma (DLBCL) [21,22]. This evolution has also been marked by a better understanding of the molecular features of these lymphomas and the improvement in immunohistochemical (IHC) techniques, allowing a more reliable diagnosis, as well as a distinction from other types of lymphomas. Progress has been made at the diagnostic level with the introduction of novel endoscopic techniques allowing better detection of sometimes subtle lesions and, in some cases, even an in vivo diagnosis of GML per se [23,24,25,26]. Finally, the association between GML and gastric adenocarcinoma has been better documented, leading to the understanding of the importance of appropriate endoscopic surveillance of these patients, implicating the crucial role of a Gastroenterologist–Endoscopist in their management. Accordingly, the aim of this review was to present, in a simple and comprehensive manner, the current status and recent advances in the field of clinical management of GML with special emphasis on gastroenterological aspects and the role of gastroenterologists in the management of these patients.

## 2. Diagnosis

The clinical symptoms of GML are not specific, and most frequently include either a peptic ulcer disease syndrome or dyspeptic symptoms [27,28]. In a French population-based EPIMALT study describing the ancient management of GML in France in real-life (outside clinical studies), including all the patients diagnosed with GML recorded within the National Cancer Registry between 2002 and 2010, the majority of the 416 patients (69%) presented either with ulcer-like syndrome or dyspepsia [27]. In rare cases, the disease may be manifested by complications like perforation or digestive bleeding [27]. It should also be stated that in a small but non-negligible proportion of patients (6%), these lymphomas may be diagnosed fortuitously during an endoscopy performed for reasons apparently not related to this disease, like screening endoscopy in asymptomatic patients, systematic work-up before bariatric surgery, anemia, or other symptoms unlikely related to lymphoma like gastroesophageal reflux disease [27]. This raises an important issue regarding the role of the Gastroenterologist–Endoscopist in the diagnosis of these lymphomas, which may be diagnosed during an upper endoscopy performed for any other reason, or even on the basis of histopathological analysis of systematic gastric biopsies obtained on a normally looking gastric mucosa.

### 2.1. Endoscopy

The endoscopic appearance of GML is not specific and may vary from a normally looking gastric mucosa, through spotty redness, large thickened gastric folds, nodular gastritis-like appearance, and rarely more pronounced lesions like ulcers or tumors [25,29,30,31,32] (Figure 1).

In a number of patients, GML is diagnosed histologically on systematic biopsies obtained on a normally looking or showing only slight abnormalities gastric mucosa [27,33,34]. In an analysis of pooled data regarding 2000 patients included in 38 studies published before 2010, GML was diagnosed on a normal or slightly hyperemic gastric mucosa in 140 patients (8.4%) [35]. This raises the question of the necessity of these biopsies, which should always be encouraged in patients undergoing upper endoscopy for any reason.

Virtual chromoendoscopy with narrow-band imaging (NBI) or blue light imaging (BLI) has a potential interest in the diagnosis of GML by allowing, in particular, better visualization of the vascular pattern of the mucosa. Indeed, NBI has been shown to be useful in GML diagnosis by allowing visualization of a unique vascular pattern specific for GML, consisting of a change of the structure of the vessels with the “tree-like appearance” organized as the trunk of the tree and its branches. It also allows us to better visualize the modified, enlarged vessels typical for GML [31,36] (Figure 2).

Another point is the possibility of in vivo diagnosis of GML, using the different methods of magnifying endoscopy, like probe-based confocal laser endomicroscopy (pCLE) or endocytoscopy, allowing in vivo the observation of gastric mucosa at a microscopic level (magnification ×1000), the first being realized after intravenous injection of fluorescein and the second after local application of methylene blue. The first report on using pCLE in a patient with GML was published in 2015 [37]. It showed that pCLE, used in association with NBI, allowed characterizing specific GML endoscopic features, like discoloration of the mucosa, absence of glandular structures, and the presence of abnormal blood vessels. A retrospective observational study including 323 patients, among whom five had GML, indicated the overall capacity of pCLE to identify any gastric lesions with a sensitivity of 72.4% and a specificity of 93.1%. In particular, it pointed to its capacity to diagnose gastric lymphomas by showing the affected mucosa densely infiltrated with identical round-shaped abnormal cells. In this study, pCLE correctly established diagnosis, confirmed by histology, of one diffuse large B cell lymphoma and five GML following pCLE criteria [25]. In a prospective pilot study including 24 consecutive patients, 62% had a positive diagnosis of GML based on histology, and the sensitivity of pCLE for lymphoma detection was 93% with 100% specificity [23]. Endocytoscopy also showed a potential to identify some specific histological features of gastric mucosa, like intra- and inter-glandular aggregation of cellular components, typical for different types of lymphomas and indicating the presence of lymphoepithelial lesions, observed in GML and associated DLBCL [38,39]. These data indicate that both techniques, endocytoscopy and pCLE, allow the identification of several histological features of GML. However, the capacity of pCLE to distinguish between GML and other types of small B cell lymphomas, like follicular lymphomas or mantle cell lymphomas, although extremely rare in the stomach, remains to be shown. Altogether, larger studies based on a higher number of patients with different types of lymphoma would be necessary to validate the diagnostic performances of pCLE in GML.

Additionally, in some cases of difficult diagnosis based on gastric biopsies, endoscopy may improve the performances of histological diagnosis by allowing the provision of larger gastric mucosa specimens obtained by endoscopic submucosal resection [40,41].

Overall, GML represents several challenges for endoscopists: the first is an initial detection requiring a precise evaluation of the entire gastric mucosa; the second is a correct diagnosis requiring several biopsies and sometimes larger gastric mucosa specimens for histological analysis; and the third is an accurate control endoscopy after the treatment, to appreciate the evolution of macroscopic lesions. Moreover, after suspicion of lymphoma on initial gastric biopsies, another confirmatory endoscopy may be required, with a sufficient number of biopsies to confirm the diagnosis and to perform gastric mapping (if not performed at initial endoscopy). It may be combined with an endoscopic ultrasound examination, which is a routine procedure in lymphoma staging [8,34,42].

However, the backbone of GML diagnosis remains histology, and it is important to highlight that a sufficient number of biopsies (at least 10) from the lesions and outside des lesions is necessary for a reliable diagnosis (histology, IHC, and detection of *H. pylori*) [8,43] and for the evaluation of the remaining gastric mucosa in search of associated gastric precancerous lesions (see Section 5 below).

### 2.2. Histopathological Features

Despite all the progress in endoscopic techniques, the histopathological analysis of gastric biopsies remains the corner stone of the GML diagnosis. This diagnosis may be challenging, as shown in the EPIMALT study, including an ancient series of 423 patients with presumed GML (but collected between 2002 and 2010, thus before the publication of guidelines), in which there were, in fact, seven patients with a false diagnosis (other types of lymphomas) and forty-four patients with a high-grade component (DLBCL) not recognized at the initial examination. This diagnosis has improved considerably during the last two decades due to the development of new IHC methods and specialized pathology networks (in France: LYMPHOPATH Network), and the obligation of a second analysis by an expert pathologist, recommended by all international guidelines [8,9,11].

A correct histopathological analysis now allows the elimination of differential diagnoses, such as chronic gastritis that preceded lymphoma, with B cell follicles surrounded by mantle cells without an identifiable marginal zone (acquired MALT) that may be florid and difficult to distinguish from MALT lymphoma, or other types of B cell lymphomas, small or large B cell lymphomas (especially resulting from the transformation of MALT lymphoma) [2,44,45]. From a morphological point of view, MALT lymphoma recapitulates the histology of Peyer’s patches. The neoplastic B cells infiltrate around the B follicles in a marginal-zone distribution to form confluent sheets of neoplastic cells that can finally overrun follicles. Importantly, the glandular epithelium is often invaded by some aggregates of neoplastic cells forming lymphoepithelial lesions. The typical neoplastic cells, called centrocyte-like because of their resemblance to germinal center centrocytes, have a small- to medium-size and slightly irregular nuclei with pale cytoplasm, admixed with some scattered large cells resembling centroblasts (Figure 3).

Plasma cell differentiation is present in up to 33% of cases [8,45]. The typical phenotype of neoplastic B cells is CD20+, BCL2+, CD10−, BCL6−, CD5−, cyclin D1−, SOX-11−, CD23−, immunoglobulin (Ig) D−, with a low ki67 proliferation index [10,42,46]. CD10 and BCL6, the two germinal center markers, could highlight residual B cells of colonized normal germinal centers (Figure 3e, CD10). Furthermore, like a subset of normal memory/marginal-zone B cells, tumor cells are typically IgM+ and IgD−. Interestingly, as it is sometimes difficult to differentiate GML from chronic gastritis or reactive lymphoid hyperplasia, a new immunohistochemical biomarker IRTA1 (Immunoglobulin superfamily Receptor Translocation-Associated 1) can be used, as this receptor can be expressed in nearly 43% of GML [47]. Additionally, a recent study suggested that BCL10 and NF-κB are useful immunohistochemical markers to predict antibiotic unresponsiveness [48]. PCR clonality may be performed to differentiate chronic gastritis from MALT lymphoma, a B cell clone favoring the latter diagnosis, while some studies, however, report clonal results in *H. pylori* gastritis.

Fluorescent in situ hybridization (FISH) or real-time (RT)-PCR can also be useful for the detection of the t(11;18)(q21;q21) translocation, the most frequent translocation found in GML (30%). Interestingly, cases with this translocation rarely undergo a high-grade transformation, but are associated with a lack of response to *H. pylori* eradication, and this translocation, according to some recent data, may also be a predictive factor of poor progression-free survival [48,49,50,51,52,53].

For the surveillance after *H. pylori* eradication, the histopathological score GELA should be used to guide the further treatment strategy [8,46].

### 2.3. Search for H. pylori

The *H. pylori* status of the patients has to be correctly established, as it is a key prognostic factor of response to eradication therapy. Histology is the principal method since, due to the high number of gastric biopsies usually available, its sensitivity is very good in this context. If histology is negative using common stains, immunohistochemistry, which further increases its sensitivity, may be additionally performed.

However, due to the risk of false negative results by histology, related to the risk of sampling error, even increased in patients receiving the treatment with proton pump inhibitors (PPI), in the absence of *H. pylori* on biopsy specimens, additional *H. pylori* tests are recommended, especially RT-PCR on gastric biopsies and, if not available, the ^13^C- Urea breath test, and/or stool antigen test [48,54,55,56,57,58,59,60]. Indeed, an RT-PCR can now be performed in any laboratory following the COVID pandemic, during which an enormous number of such tests have been performed, and kits for *H. pylori* are commercially available, at least in Europe. This test has the best accuracy and has improved the *H. pylori* diagnosis during the last years.

Serology is an indirect test, not submitted to sampling error, which can be positive even if the patient is receiving PPI treatment, and also has the advantage of remaining positive for months after the eradication of *H. pylori*. A positive *H. pylori* status is defined as positive to at least one of the methods, histology, RT-PCR, or serology, considered the most efficient in the context of GML [8,43,61].

The negative *H. pylori* status may be declared only if all the techniques, including serology, are negative, and is more reliable if several methods are used [43,48].

### 2.4. The Case of H. pylori-Negative Lymphomas

Another clinical challenge in GML is represented by *H. pylori*-negative GML. A recent excellent review has been specifically devoted to this topic [62]. Therefore, in the present review, we will address this issue very briefly.

The reported incidence of *H. pylori*-negative GML has been increasing over the last years, and in different studies, it accounts for 10–40% of all GML cases [27,42,63,64].

In the absence of *H. pylori*, two categories of *H. pylori*-negative GML should be considered: “false” *H. pylori*-negative lymphomas and “true” *H. pylori*-negative lymphomas. The increase of false *H. pylori*-negative GML may be partly related to an increased usage of PPI treatment. To rule out the false negatives, all detection methods should be reviewed, and *H. pylori* negativity should be ascertained by confirming *H. pylori*-negative serology and RT-PCR [63,65]. Indeed, a positive serology may indicate a past, treated infection, the case which does not really correspond to the “true” *H. pylori*-naïve GML.

Among the “true” *H. pylori*-negative GML, there are those associated with non-*H. pylori* Helicobacters (NHPH) species, and those, less well defined, are associated with other bacteria, non-bacterial antigens, or some autoimmune diseases like Hashimoto thyroiditis [66].

In the case of “true” *H. pylori*-negative GML, other *Helicobacter* species should be considered and searched for [29,67,68,69,70], in particular, because the presence of these other *Helicobacter* species seems to be a predictive marker of a better response to *H. pylori* eradication treatment as compared to the patients without any *Helicobacter* [71]. The diagnostic methods for NHPH are still being discussed, and no test has been validated so far to rule out NHPH infection in patients with *H. pylori*-negative GML. However, different methods may be considered, in particular molecular, like amplification and sequencing of 16S rRNA gene of *Helicobacter* species or a metagenomic approach.

Moreover, patients with *H. pylori*-negative GML show different gastric microbiota composition as compared to control patients, including the genera *Burkholderia*, known to be an opportunistic pathogen, capable of promoting the growth of follicular lymphoma, and *Sphingomonas*, capable of activating NK-cells and systemic immunity, found to be more abundant, while *Prevotella* and *Veillonella*, known as suppressors of inflammation, were found to be less abundant [72]. However, at this stage, no consistent evidence exists allowing to confirm a causative role of microbiota diversity in GML nor its therapeutic implications [63,73].

In terms of treatment, the same treatment protocols used for *H. pylori* may be used for other Helicobacters.

## 3. Pre-Treatment Work-Up and Staging

### 3.1. Pre-Treatment Diagnostic Measures

The standard pre-treatment work up in GML, recommended by the majority of guidelines, includes a medical history and physical examination, ear–nose–throat examination, search for enlarged lymph nodes and hepatosplenomegaly, thoraco–abdomino–pelvic CT scan, upper and lower endoscopy with gastric endoscopic ultrasound, and blood tests, including a complete blood count, renal and liver function tests, protein electrophoresis, lactate dehydrogenase (LDH), beta-2 microglobulin levels (B2M), serum and urine immunofixation, and viral serologies (for hepatitis A, B, C viruses, human immunodeficiency virus, Epstein–Barr virus, and cytomegalovirus). A bone marrow biopsy is not recommended in routine, in the absence of any sign of dissemination, given its low rate of positive results and absence of clinical impact [10,74,75].

The interest in ^18^F-FDG-CT-Scan is still debated. Indeed, MALT lymphomas are generally classified among the ‘non-FDG avid’ lymphomas based on heterogeneous and contradictory results of several studies [76,77], and currently, ^18^F-FDG-CT-Scan is not recommended systematically by any of the guidelines. However, its interest in GML is being evaluated in ongoing studies and, in particular, its capacity to predict the transformation into DLBCL [78,79].

More recently, other new imaging techniques have been proposed, such as positron emission tomography/magnetic resonance imaging (PET/MRI), which might be useful in GML, mostly for the assessment of disease extent and response to treatment [80].

### 3.2. Staging Systems or Classifications

The pre-treatment work up should allow for the precise determination of the GML stage, which may be expressed using the Ann Arbor modified by Mushoff classification [81,82] or the Paris classification [83]. The latter is the best adapted to GML and should be recommended. The Paris classification, based on endoscopic ultrasound, adapts the TNM system used for other epithelial cancers of the digestive tract to GML, and it allows to better describe the depth of invasion in the gastric wall (T), which is an important predictive factor of response to *H. pylori* eradication. Indeed, the tumors invading the muscular layer (stage T2) are less likely to respond to eradication treatment [42,84].

Most of the GML (>90%) are localized and of stage I or II, with very few at more advanced stages [84,85].

## 4. Treatments

### 4.1. H. pylori Eradication

The first line of treatment in all cases, independently of *H. pylori* status and of the disease stage, is an *H. pylori* eradication therapy. This attitude is adopted by most of the European guidelines [8,9,10,11]; however, the more recently published NCCN guidelines recommend in *H. pylori*-negative patients the first line of treatment by radiotherapy, or, if contraindicated, by immunotherapy with Rituximab [86].

Different eradication regimens may be proposed; they are all based on the association of two or three antibiotics, or antibiotics and bismuth salts, with a double dose of PPI. The optimal duration of treatment is now established at 14 days, with the exception of Pylera^®^ (association of bismuth subcitrate + metronidazole + tetracycline), which is administered together with omeprazole for 10 days. The recent Maastricht consensus recommends proposing the first-line treatment according to the results of antimicrobial susceptibility testing, which should be relatively easy to perform in the case of GML given the fact that several gastric biopsies are usually obtained (>10), allowing to reserve two biopsies for the antibiogram. This antibiogram can be performed on gastric biopsies after bacterial culture, which allows testing all antibiotics of potential interest, or using an RT-PCR which allows the detection of *H. pylori* and the mutations leading to clarithromycin resistance. In particular, the detection of clarithromycin resistance by RT-PCR must be encouraged since, in case of a strain susceptible to clarithromycin, an optimized triple therapy (amoxicillin 3 × 1 g + clarithromycin 2 × 500 mg + esomeprazole 3 × 40 mg/day for 14 days) may be proposed. This regimen may be particularly advantageous in the case of GML given that some data are suggesting some anti-lymphoma activity of clarithromycin per se, and its efficacy in extra nodal MALT lymphomas [87,88]. If antimicrobial susceptibility testing is not feasible, or the bacterial strain is resistant to clarithromycin, the first-line empirical treatment by Pylera^®^ should be proposed. The efficacy of the eradication treatment must always be verified, the best by the urea breath test performed at least 6 weeks after cessation of treatment. In case of its positivity, second-line treatment should be proposed, always in accordance with antibiotics testing results, which requires a new upper endoscopy, indicated otherwise, to verify the evolution of macroscopic lesions [8,58].

The response to treatment is evaluated endoscopically by showing a regression of macroscopic lesions (i.e., ulcers), if present initially, at control endoscopy, and histologically, showing a histological regression of lymphocytic infiltration of gastric mucosa, the best evaluated according to the GELA classification [46].

After successful eradication, in the case of *H. pylori*-positive GML, complete remission is reported in 70% to 80% of cases, being higher for stage IE as compared to stage IIE [10,15,42,89,90,91].

The initial studies showed a lymphoma regression rate after successful eradication up to 90%, while in the more recent studies, this regression rate decreased to 50–60%, all stages included, probably due to a higher proportion of *H. pylori*-negative GML and variable stages of the disease not specified in all the studies [89,92]. In the case of *H. pylori*-negative GML, this rate is much lower, but still, complete remission may be achieved in 15 to 30% of patients [93,94]. The possible explanation may be a false *H. pylori*-negative status or the involvement of other bacteria (*Helicobacter* or non-*Helicobacter*) susceptible to the antibiotic treatment. A recent meta-analysis of 25 studies reported an overall complete remission rate in *H. pylori*-negative GML of 29.3%. Notably, the rate was slightly lower (27.9%) in studies where the *H. pylori* status was evaluated by serology, indicating a higher incidence of false negative results obtained by other testing procedures if serology is not included [65]. In a series of 137 patients with GML, response to *H. pylori* eradication treatment was observed in 75% of *H. pylori*-positive patients and in 29% of the *H. pylori*-negative group [95].

Apart from the negative *H. pylori* status [7,71], other risk factors of lack of response to eradication treatment include the stage and the depth of gastric wall invasion as evaluated by endoscopic ultrasonography [42], the regional lymph node involvement [42], and the presence of the chromosomal t11;18 translocation which results in deregulation of MALT1 or Bcl-10 [52,53], as well as other genetic features, such as overexpression of miR-142-5p and miR-155 [96]. The translocation t(11;18)/API2-MALT1 is reported in 7 to 25% of GML cases, often in more advanced stages (IIE or above) [71,97]. However, among GML non-responding to *H. pylori* eradication, this translocation was detected in 47% and 68% of GML at stage IE and stage IIE or above, respectively, in multicenter retrospective studies [8,52,53]. Wundisch et al. showed that only 15% of GMLs were t(11;18) positive, but that both t(11;18) and ongoing monoclonality were associated with a significantly higher risk for lack of response to *H. pylori* eradication or relapse [98]. Moreover, a strong BCL10 nuclear expression in GML has been associated with a non-response to *H. pylori* eradication [99].

Nevertheless, most of the current guidelines, especially in Europe, recommend *H. pylori* eradication for all GML patients, independently of the disease stage, the translocation status, or *H. pylori* positivity [8,9]. However, it should be noted that the NCCN guidelines recommend in *H. pylori*-positive cases with t(11;18), already at the first line of treatment, together with *H. pylori* eradication, radiotherapy or, if contraindicated, immunotherapy with Rituximab (NCCN guidelines) [86].

Despite NHPH-associated GML being an extremely rare clinical entity, eradication therapy seems to be also the optimal first-line therapeutic choice leading more often to a complete remission than in non-*H. pylori* GML (12/16, 75% vs. 3/13, 23%) [68,71].

### 4.2. Surveillance after Effective H. pylori Eradication

After successful eradication, it is mandatory to have at least two negative control examinations at one-year intervals to confirm GML regression using the GELA criteria [46]. According to these criteria, a complete lymphoma regression can be declared either in case of a total disappearance of the lymphoid infiltrate (complete histological response, CR) or in case of persistence of only small lymphoid aggregates defined as a probable minimal residual disease (pMRD), which must be associated with a disappearance of any macroscopic, endoscopically visible lesions. If the features of partial histological response (rRD) or no change (NC) are present, a non-regression of lymphoma is declared. It is important to wait a sufficiently long time before declaring a non-regression in order to avoid the risk of over-treatment. In an international series of 108 patients with GML followed-up after *H. pylori* eradication for a mean period of 42.2 months, in 62% of these patients, a residual lymphoma infiltrate was still present after 12 months, and in 55% after 24 months following *H. pylori* eradication [22]. Accordingly, most of the guidelines suggest a 24-month observation period before considering other treatments, but it has been shown that in some cases, the disappearance of lymphoma lesions may be observed even later, after several years [8,12,22,27,89]. In the French EPIMALT study, in one patient, complete remission was achieved only 68 months after eradication [100].

Several studies addressed the question of a possible “watch and wait” strategy in patients with persisting residual disease after *H. pylori* eradication, and they showed a very good long-term outcome and a low risk of transformation into a high-grade lymphoma in these patients [22,84,98,101]. In an international series of 108 patients with GML followed-up after *H. pylori* eradication for a mean period of 42.2 months, 102 (94%) had a favorable course of the disease, the progression of lymphoma was seen only in four patients, and only one developed a high-grade lymphoma [22]. In a prospective multicenter trial, including 120 patients with stage EI1 GML treated by *H. pylori* eradication and followed for a median period of 122 months, from whom 80% achieved complete remission and 17% presented a histological residual disease, no progression was observed in any of the patients, and all but one achieved a second complete remission [102]. An excellent long-term clinical outcome of GML after *H. pylori* eradication was confirmed in a large-scale Japanese multicenter study of 323 patients with GML followed-up for a median period of 6 years (min–max: 3–14.6 years), in which the strict GELA criteria were used to evaluate histological response to eradication treatment. In this study, complete remission was achieved in 77% of patients after *H. pylori* eradication, and the lymphoma relapse occurred in only 3.1% of patients. From 97 non-responders, progressive disease was observed in 27 patients who subsequently received other treatments, while 14 out of 70 non-responders without progressive disease were submitted to the “watch and wait” strategy. The overall 10-year survival of all the patients was 95% [89].

All these data show a very good long-term prognosis in these patients, justifying a “watch and wait” strategy, provided that a regular endoscopic follow-up can be offered with at least annual upper endoscopy with multiple gastric biopsies.

### 4.3. Other Treatments after H. pylori Eradication

In case of persistence of GML after *H. pylori* eradication, and if the “watch and wait” strategy is not considered the best option, a second-line treatment by a low to moderate dose (25–40 Gy) involved site radiation therapy (ISRT) in this usually localized disease may be proposed. As reported in several series, it gives excellent results with an almost 100% complete remission rate and very few relapses (5 to 10%), with an excellent 5-year overall survival of over 95% in different series [17,19,103,104,105,106,107,108]. It was shown to be well tolerated with no significant late toxicities [109], and a very good long-term outcome [105].

Even reduced doses of 24 and 25.2 Gy proved to be safe and effective, leading to high complete remission rates [19,108]. In the recently published retrospective series of 33 patients treated with radiotherapy with a median follow-up of 66 months, the local relapse-free survival at 5 years was 92%, and there were no significant late toxicities or treatment-related death [109].

The efforts to decrease the dose of radiation applied continue, and in a currently ongoing international multicenter trial, “GDL-ISRT 20 Gy”, initiated by the International Lymphoma Radiation Group and the German Lymphoma Alliance in 2019, the use of ISRT with only 20 Gy in GML is evaluated [110].

In parallel to the radiation dose reduction, attempts to reduce the target volume have been undertaken, with the aim of reducing the toxicity and improving the tolerability of modern radiotherapy in GML [111].

Radical surgery is no longer recommended, as it has not shown superiority over irradiation and may result in local complications and reduced quality of life [8,28]. It is reserved for rare GML complications, such as bleeding or perforation.

In very rare cases of disseminated diseases or with lymph node involvement, a systemic treatment based on immunochemotherapy may be proposed, usually led by Hematologists [28]. Different regimens have been evaluated, mostly in disseminated extra nodular MALT lymphoma and more rarely in localized GML [112]. The association of Rituximab and chlorambucil appeared more effective than chlorambucil alone in a phase III study [113]. More recently, the association of rituximab with bendamustine has been reported as giving a complete remission already after three cycles [20]. It should be kept in mind, however, that in localized disease, systemic treatment may be unnecessary and even toxic, as demonstrated by some studies showing secondary neoplastic complications of alkylated agents [114,115,116].

A possibility of endoscopic treatment of superficial GML lesions has also been suggested, based on several reports showing that such a treatment (namely by endoscopic submucosal dissection), applied in *H. pylori*-negative GML or after *H. pylori* eradication, is feasible and may be effective [117,118,119]. The 5-year overall survival in patients who had never received *H. pylori* eradication treatment, but only other treatments, was slightly inferior compared to those who received *H. pylori* eradication treatment followed by other treatments in case of lymphoma persistence (78.5% and 84.3%, respectively) [27].

## 5. GML, Gastric Precancerous Lesions and Gastric Adenocarcinoma

An important issue in patients with GML is the necessity to search for the presence of associated gastric precancerous lesions: atrophic gastritis (AG), intestinal metaplasia (IM), and dysplasia within and outside the lymphoma lesions, both at initial examination and during the follow-up. Indeed, an increased risk of adenocarcinoma in patients with GML has been well documented and estimated at 6 to 10 times fold as compared to the general population [102,120,121]. In the German study by Wündisch et al., a gastric adenocarcinoma developed in 5 out of 96 patients in complete remission of GML during the period of 33 to 103 months after diagnosis of GML [102].

Several cases of concomitant GML and gastric adenocarcinoma have also been reported [122,123,124,125,126].

Furthermore, the association between GML and gastric carcinogenesis is also supported by data coming from more fundamental studies, showing that both processes (lymphomagenesis and carcinogenesis) share some common mechanisms, and in particular, the lymphocyte T stimulation, induced by *H. pylori*, the same pathogen involved in both GML and gastric adenocarcinoma pathogenesis, with an excessive TH1 pro-inflammatory immune response likely responsible for the evolution towards these two diseases [127,128].

The association between GML and gastric precancerous lesions also seems well-established based on several studies showing that these lesions are frequently present in patients with GML. In the French EPIMALT study, IM was found in 33% of the patients with GML [27]. In a retrospective study of 179 patients with GML, AG and/or IM were found in 46% of the patients [129], while these lesions were present in 68% of 32 patients with GML identified in the Dutch histopathology registry [130]. Thus, the proportion of GML patients with gastric precancerous lesions appears significantly higher than the one observed in non-lymphoma European patients undergoing upper endoscopy for any reason, which varies from 3% to 22% [131,132]. Several studies also suggest that these lesions may progress more rapidly in the context of GML. A retrospective analysis of the data of 50 patients with gastric lymphomas, including 40 patients with GML and 10 with DLBCL, showed the presence of IM in 24% of the patients at diagnosis and the appearance of IM in 57.9% during the median follow-up of 30 months [133]. The overall progression of IM was observed in 21.2% of cases, and this progression rate appears higher than the one observed in non-lymphoma patients (from 4 to 14%) [134,135]. In a study including 45 patients with GML, the progression of AG and IM during a median follow-up period of 54 months was significantly higher than in age-matched non-ulcer dyspepsia patients without lymphoma [136]. In another study, out of 70 patients with GML having received conservative treatment (*H. pylori* eradication +/− chemotherapy with an alkylating agent) and being considered in complete remission during endoscopic surveillance, an early gastric adenocarcinoma was diagnosed after a mean follow-up period of 9.5 years in four patients, at the same location as the lymphoma. All these patients underwent surgery, and histopathological examination of gastrectomy specimens showed a residual GML in all of them. Out of these four patients, only one presented IM at the initial GML diagnosis; however, in three of them, IM and dysplasia were present at the time of gastric cancer diagnosis, suggesting that these lesions appeared in the meantime [137].

All these data strongly support the link between GML and gastric adenocarcinoma and the necessity of strict surveillance of the gastric mucosa after GML treatment, even in cases of apparent complete remission, especially if the precancerous lesions are present concomitantly with the lymphoma, but also in the absence of these lesions which may appear afterward. This highlights the necessity of appropriate endoscopic surveillance of these patients and the central role of the Gastroenterologist in this surveillance. The modalities of this surveillance are not well standardized since this particular situation has not been addressed in the different international guidelines on the management of gastric precancerous lesions published so far [138,139,140]. However, it appears reasonable to propose a good quality upper endoscopy with systematic gastric biopsies (at least two from the antrum and two from the corpus) for histological evaluation according to the OLGA/OLGIM system, every 3 to 5 years, with the time interval adapted taking into account additional factors (presence and severity of gastric precancerous lesions, family history of gastric carcinoma, ethnic origin, persistence of *H. pylori* infection, age, comorbidities, etc.) [138,141]. In case of the absence of any gastric mucosa lesions after lymphoma remission, it may be reasonable to propose to pursue the surveillance still for 10 years and to stop it if the gastric mucosa remains normal; however, dedicated studies would be necessary to validate this strategy.

## 6. Surveillance after Complete Remission of Lymphoma

The complete regression of GML may be achieved either by *H. pylori* eradication only or after additional treatments following eradication (radiotherapy or immune-chemotherapy).

Even in complete remission, surveillance is required due to the increased risk of gastric adenocarcinoma, particularly if precancerous lesions are present, despite the low risk of lymphoma relapse (see Section 5).

## 7. Prognosis

The overall prognosis in GML is good; the 5-year overall survival overcomes 90% in most of the studies, and is very good even in more advanced (disseminated) stages of the disease [27,84,98,142].

The major risk is the transformation into high-grade lymphoma, which is very rare, evaluated in different follow-up studies at 1 to 3% [22,89,143,144]. In a French prospective series of 53 patients with stage IE or IIE GML persistent after *H. pylori* eradication, followed-up for a mean period of 7.6 years, complete remission was achieved in all but one patient, and the disease-specific and overall 5-year survival was 94% and 100%, respectively [17].

In a Japanese cohort of 98 patients with MALT lymphoma, including 52 patients with GML, the 3-year overall survival for the entire cohort was 100%. The 3-year progression-free survival in GML patients was better in those with localized disease (100%) as compared to those with disseminated disease (76%) [145].

Altogether, these arguments are in favor of a non-aggressive attitude in GML, based on *H. pylori* eradication treatment, long “watch and wait” strategy, local low-grade radiotherapy in localized, persistent disease, and systemic treatment only reserved for progressive and disseminated disease at high risk of transformation [143,144].

## 8. Conclusions

GML are rare gastric tumors, essentially related to a chronic infection with *H. pylori*, rarely to other *Helicobacter* species, with a usually indolent course and good prognosis, which require specific management by a Gastroenterologist within a multidisciplinary team to avoid excessive treatment. Special attention must be paid to the risk of gastric adenocarcinoma, and dedicated guidelines are warranted. New advances are expected in the field of the endoscopic in vivo diagnosis of these lymphomas, and better clinical management based on our knowledge of their physiopathology and evolution.

## Figures and Tables

**Figure 1 cancers-15-03811-f001:**
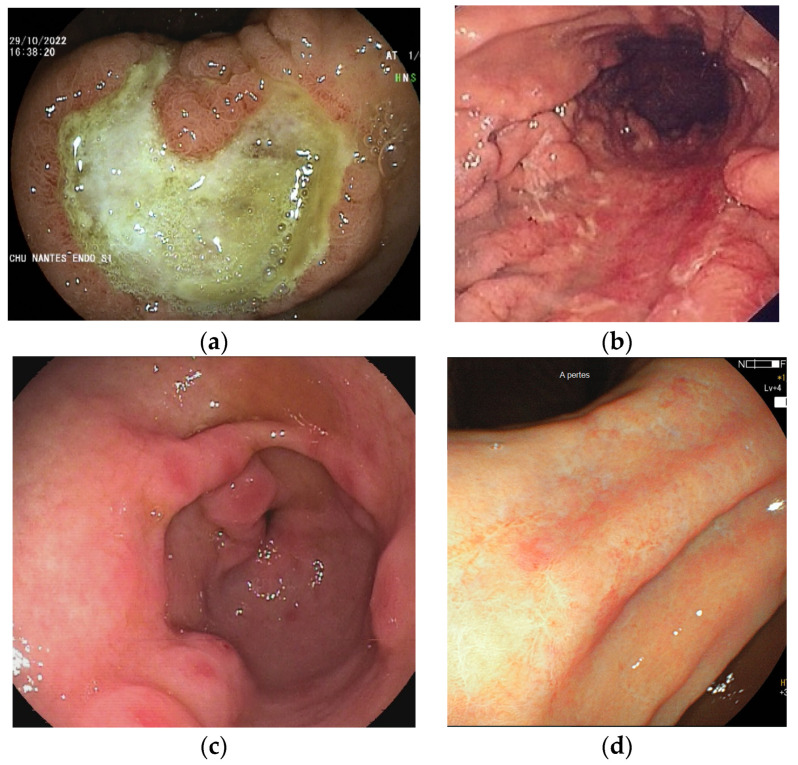
Different endoscopic aspects of gastric MALT lymphoma. (**a**) Ulcer; (**b**) enlarged gastric folds with erythema; (**c**) erosive nodules; and (**d**) erythematous scar. Images: Courtesy of Dr. M. Collins and Dr. N. Musquer, Nantes, France.

**Figure 2 cancers-15-03811-f002:**
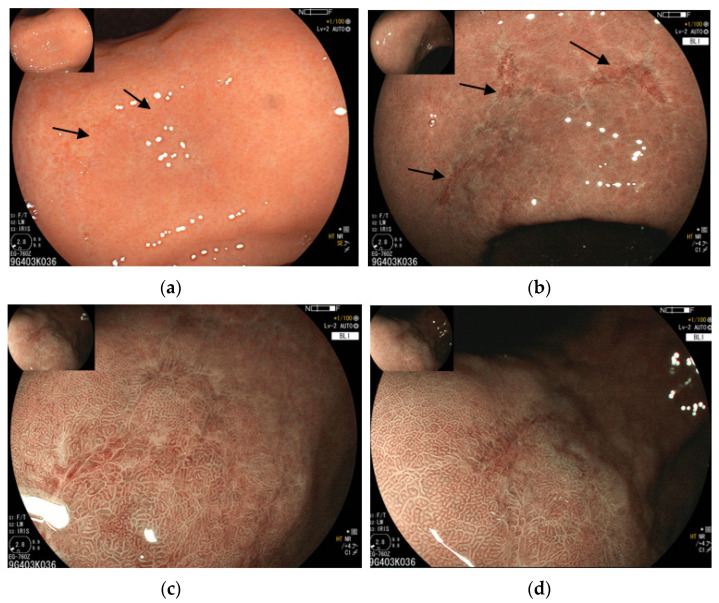
Endoscopic aspects of gastric MALT lymphoma by virtual chromoendoscopy. (**a**) White light endoscopy: only very slight lesions are visible (indicated by arrows); (**b**) endoscopic image in virtual chromoendoscopy by BLI (blue light imaging) and magnification: the lesions are much more visible, with clearly abnormal vascular patterns, showing a “tree-like appearance” with abnormal vessels organized as a trunk of the tree with its branches, and linear erosions; and (**c**,**d**) the same images in BLI and magnification, focused on linear erosions. Images: Courtesy of Dr G. Velut, Nantes, France.

**Figure 3 cancers-15-03811-f003:**
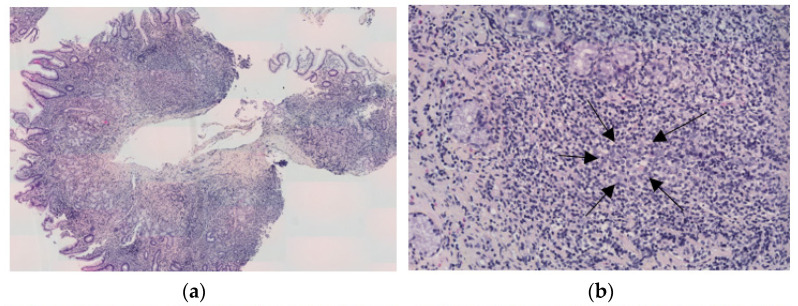
Histopathological features of gastric MALT lymphoma. (**a**) Large infiltration of gastric mucosa by a proliferation of lymphoid cells with a nodular pattern (HES, ×5 magnification); (**b**) some nodules contain a colonized residual germinal center (arrows, HES ×20 magnification); (**c**) tumor cells have small- to medium-sized and irregular nuclei with mature chromatin and colonized glandular epithelium to form lymphoepithelial lesions (arrow, HES ×40); (**d**) tumor cells express diffusely CD20. This immunostaining highlights lymphoepithelial lesions (arrows, ×20 magnification); (**e**,**f**) tumor cells are CD10- (**e**) and CD23- (**f**). CD10 highlights residual germinal centers, and CD23 shows residual follicular dendritic cells meshwork within tumor nodules (×5 magnification); (**g**) tumor cells are CD5-, associated with some reactive CD5+ T cells around the tumor nodules (×5 magnification); and (**h**) MALT lymphoma is associated with *Helicobacter pylori*, present within the gastric glands (arrow, HP immunostaining, ×40 magnification).

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
