# Peer review of "Clinical Management of Patients with Gastric MALT Lymphoma: A Gastroenterologist’s Point of View"

_cancers, 2023, doi:10.3390/cancers15153811_

Round 1

Reviewer 1 Report

This is the well-written manuscript about clinicopathological features of gastric MALT lymphoma. However, I have some concerns about this review article.

Major comments:

1.      In line 105-107, the authors describe that NBI is useful way to diagnose gastric MALT lymphoma (gMALT lymphoma). Therefore, I think they should explain the findings of NBI in gMALT lymphoma more detail and add endoscopic images of that to Fig 1.

2.      In line 120-122, the authors show that pCLE could distinguish DLBCL and MALT lymphoma in stomach. However, I don’t think it could exclude the possibility of Mantle cell lymphoma and Follicular lymphoma, even they are rare disease. How the possibility of them is excluded?

3.      The authors described that “the backbone of GML diagnosis remains histology, and it is important to highlight that a sufficient number of biopsies (at least 10) from”.  However, I think that gMALT lymphoma could be diagnosed by less than 10 biopsy specimens. In addition, H. pylori infection could be evaluated by other examinations except for biopsy. The authors should address this concern.

4.      In Fig2, authors need to post higher magnification of HE stained image to recognize small- to medium-size and irregular nuclei. In addition, I recommend that they should change to higher quality images of immunohistochemical staining, especially CD20.

5.      In line 195, authors should described the pattern of BCL2, BCL6, IgM, and IRTA1.

6.      In the paragraph of 7 (Prognosis), I recommend that authors should describe the rate of 5-year OS in patients with gMALT lymphoma in disseminated stages and the rate of PFS.

Minor comments:

1.      In line 195, “cyclin D1” “cyclin D1-”

2.      Authors describe the t(11;18)(p21;p21) translocation in line 202-203, I think it is mistake of t(11;18)(q21;q21).

3.      Authors should check abbreviations again. (gastric MALT lymphoma or GML)

Author Response

RESPONSES TO THE REVIEWERS’ COMMENTS

We wish to thank the Editor and the Reviewers for your valuable comments, please find enclosed the point-to-point response to all the comments. All modified and added text to the manuscript appears in bold and yellow.  

REVIEWER 1:

This is the well-written manuscript about clinicopathological features of gastric MALT lymphoma. However, I have some concerns about this review article.

Major comments:

  1. In line 105-107, the authors describe that NBI is useful way to diagnose gastric MALT lymphoma (gMALT lymphoma). Therefore, I think they should explain the findings of NBI in gMALT lymphoma more detail and add endoscopic images of that to Fig 1.

Response: The paragraph on virtual endoscopy (NBI and BLI, paragraph 2.1, page 3)  has been developed, according to the Reviewer’s suggestions, and additional figure (Figure 2) has been added with the BLI images of gastric MALT lymphoma.

Virtual chromoendoscopy with narrow band imaging (NBI) or blue light imaging (BLI), has a potential interest in the diagnosis of GML by allowing, in particular, a better visualization of the vascular pattern of the mucosa. Indeed, NBI has been shown to be useful in GML diagnosis by allowing to visualize an unique vascular pattern specific for GML, consisting of a change of the structure of the vessels with the “tree-like appearance”, organized as the trunk of the tree and its branches. It also allows to better visualize the modified, enlarged vessels, typical for GML.

  1. In line 120-122, the authors show that pCLE could distinguish DLBCL and MALT lymphoma in stomach. However, I don’t think it could exclude the possibility of Mantle cell lymphoma and Follicular lymphoma, even they are rare disease. How the possibility of them is excluded?

Response: We fully agree with the reviewer that the distinction between different types of small B cell lymphomas by pCLE may be difficult. Furthermore, due to the extremely low prevalence of Mantle cell and follicular lymphomas in the stomach, it would be a challenge to define strict criteria for pCLE diagnosis of these lymphomas. Besides, pCLE is performed only in some expert centers, and cannot be routinely used. Thus, today, this technique should be considered an useful, complementary tool in trained hands, but cannot replace histology that remains the gold-standard. Given all these limitations, the message in the text has been attenuated (page 5):

        However, the capacity of pCLE to distinguish between GML and other types of small B cell lymphomas, like follicular lymphoma or mantle cell lymphomas, although extremely rare in the stomach, remains to be shown. Altogether, larges studies, based on a higher number of patients with different types of lymphomas, would be necessary to validate the diagnostic performances of pCLE in GML.

  1. The authors described that “the backbone of GML diagnosis remains histology, and it is important to highlight that a sufficient number of biopsies (at least 10) from~”.  However, I think that gMALT lymphoma could be diagnosed by less than 10 biopsy specimens. In addition,  pyloriinfection could be evaluated by other examinations except for biopsy. The authors should address this concern.

Response: We agree with the Reviewer that the histopathological GML could be diagnosed by less than 10 biopsy specimens. However, it depends both on the size and on the content of biopsies (normal, gastritis or tumor), and the more tumor tissue we have, the more proper is the diagnosis of GML and allows exclusion of differential diagnosis such as chronic gastritis, or other more rare small B-cell lymphomas.

We also agree with the Reviewer that other than histology methods should be used for the diagnosis of H. pylori, especially serology but also PCR or urea breath test, all these methods have been largely presented in the corresponding paragraph in the manuscript (paragraph 2.3, page 7).

  1. In Fig2, authors need to post higher magnification of HE stained image to recognize small- to medium-size and irregular nuclei. In addition, I recommend that they should change to higher quality images of immunohistochemical staining, especially CD20.

Response: A new HE stained image at higher magnification (x400) has been added showing tumor cell with small to medium size and irregular nuclei infiltrating epithelial glands to realize lymphoepithelial lesions (Figure 3c). In addition, we also have added a higher magnification of CD20 immunostaining image (at 400 magnification) showing lymphoepithelial lesions (Figure 3 d). The legend of the Figure 3 (corresponding to the Figure 2 of the initial version) has been developed accordingly (page 6)

  1. In line 195~, authors should described the pattern of BCL2, BCL6, IgM, and IRTA1.

Response: The description of the expression patterns of these 4 proteins has been added (page 7):

CD10 and BCL6, the two germinal centers markers, could highlight residual B-cells of colonized normal germinal centers (Figure 3e, CD10). Furthermore, like a subset of normal memory/marginal zone Be cells, tumor cells are typically IgM+ and IgD-. Interestingly, as it is sometimes difficult to differentiate GML from chronic gastritis or reactive lymphoid hyperplasia, a new immunohistochemical biomarker IRTA1 (Immunoglobulin superfamily Receptor Translocation-Associated 1) can be used, as this receptor can be expressed in near 43% of GML (Ayada et al, J clin Exp Hematop 2022;62:195).

  1. In the paragraph of 7 (Prognosis), I recommend that authors should describe the rate of 5-year OS in patients with gMALT lymphoma in disseminated stages and the rate of PFS.

Response: Some more data on OS data for disseminated GML and the PFS have been added, paragraph 7, page 13-14):

In a Japanese cohort of 98 patients with MALT lymphoma, including 52 patients with GML, the 3-year overall survival for the entire cohort was of 100%. The 3-year progression free survival in GML was significantly better in patients with localized disease (100%) than in those with disseminated disease (76%). (Ueda et al, Leukemia and Lymphoma 2013).

Minor comments:

  1. In line 195, “cyclin D1” → “cyclin D1-”

Response: the error has been corrected, thank you

  1. Authors describe the t(11;18)(p21;p21) translocation in line 202-203, I think it is mistake of t(11;18)(q21;q21).

Response: This mistake has been corrected, thank you

  1. Authors should check abbreviations again. (gastric MALT lymphoma or GML)

Response: The abbreviations have been checked, and GML abbreviation has been

Author Response

RESPONSES TO THE REVIEWERS’ COMMENTS

We wish to thank the Editor and the Reviewers for your valuable comments, please find enclosed the point-to-point response to all the comments. All modified and added text to the manuscript appears in bold and yellow.  

REVIEWER 2

Issue:

The authors present an interesting review on the clinical management of patients with gastric MALT lymphoma (GML) from the gastroenterologist point of view.

This review reflects diagnostic and histopathological aspects, the importance of H. pylori status, work-up, staging systems, treatments, gastric (pre)cancerous lesions and the prognosis of GML.

Limitations

The authors also point out limitations of the cited publications.

General suggestions for improvement:

The manuscript is mostly well written. But the importance of radiotherapy should be presented and underlined in more detail.

The following corrections are required.

Further detailed corrections:

1.The authors state that the current guidelines recommend „H. pylori eradication for all GML patients, independently of the disease stage, of the translocation status or H. pylori positivity“. However in doing so they only refer to older guidelines from 2011 (EGILS) and from ESMO (2013) and thereby only referring to European guidelines.

The authors, throughout the manuscript, should supplement and address necessarily also to the NCCN Guidelines Version 5.2023: Therein for MZL of the stomach Stage I1-II1 H.p. negative is recommended the ISRT (preferred) or Rituximab (if ISRT is contraindicated).

Response : Thank you for this comment. Indeed, H. pylori eradication treatment is recommended by most of European guidelines as first line treatment in all patients with GML, even H. pylori -negative. This recommendation is based on several reports on GML regression after this treatment in 10% to 30% of H. pylori-negative patients in different series published (Zullo et al J Clin Gastroenterol 2013;47:824-7, Raderer et al Ann Hematol 2015;94:969-73, and others). This result is probably due to some false H. pylori-negative cases, but also may be due to eradication of some other bacterial species which might play a causative role in GML. In front of a potentially serious disease like GML, an eradication treatment, even if applied by excess, appears well justified in terms of a potential benefit-cost effect.

However, we agree with the Reviewer that the point of view represented by our American Colleagues, reflected by the very recently published NCCN Guidelines 2023, is important to be cited, and this comment has been added (paragraph 4.1, page 9):

This attitude is adopted by most of the European guidelines, however, the more recently published NCCN guidelines, recommend in H. pylori negative patients the first line of treatment by radiotherapy or, if contraindicated, by immunotherapy with Rituximab (REF: NCCN Guidelines Version 5.2023

2.Moreover according to the NCCN Guidelines Version 5.2023 the t(11;18) is a predictor for lack of tumor response (<5%) to antibiotics. These patients should be considered for alternative therapy to lymphoma. NCCN Guidelines Version 5.2023 for H.p. positive, t(11;18) positive in case of being asymptomatic recommend ISRT or rituximab. (see Liu H et al. Gastroenterology 2002)

The authors should add that according to the NCCN Guidelines Version 5.2023: For MZL of the stomach Stage I1-II1 H.p. negative is recommended the ISRT (preferred) or Rituximab (if ISRT is contraindicated).

Response : This comment has been added (page 10), The reference of Liu et al is already included :

Nevertheless, most of the current guidelines, especially in Europe, recommend H. pylori eradication for all GML patients, independently of the disease stage, of the translocation status or H. pylori positivity. However, it should be noted that the NCCN guidelines, recommend in H. pylori- positive cases with t(11;18), already at the first line of treatment, together with H. pylori eradication, radiotherapy or, if contraindicated, immunotherapy with Rituximab (NCC guidelines).

  1. The GML treatment strategies have evolved considerably, among others, to local low dose radiotherapy. Under 4.3. the authors state applied RT doses of 24 and 25,2 Gy.

Regarding the historical trend to decrease applied radiation doses, the authors under 4.3. should add the international multicenter trial 'GDL-ISRT 20 Gy', initiated by the ILROG and the GLA in 2019, led from the University Hospital of Münster, Germany (https://clinicaltrials.gov) (Trial number NCT04097067). This ongoing trial with participation of international centers prospectively investigates the use of ISRT with only 20 Gy in GML (and indolent lymphoma of the duodenum).

Response : This comment has been added (page 11):

The efforts to decrease the dose of radiation applied continue, and in a currently ongoing international multicenter trail “GDL-ISRT 20 Gy”, initiated by the International Lymphoma Radiation Group and the German Lymphoma Alliance in 2019, the use of ISRT with only 20 Gy in GML is evaluated.  

4.Even earlier than the historical trend to dose reduction, a radiation field decrease was successfully investigated prospectively in the multicenter studies by the German Study Group for Gastrointestinal Lymphoma.

The authors should definitely add the development of the target volume reduction in the manuscript and cite the corresponding study results of the German multicenter studies, which are of great importance for the low-toxicity tolerability of modern radiotherapy in GML (Reinartz G, Pyra RP, Lenz G, Liersch R, Stüben G, Micke O, Willborn K, Hess CF, Probst A, Fietkau R, Jany R, Schultze J, Rübe C, Hirt C, Fischbach W, Bentz M, Daum S, Pott C, Tiemann M, Möller P, Neubauer A, Wilhelm M, Willich N, Berdel WE, Eich HT. Favorable radiation field decrease in gastric marginal zone lymphoma : Experience of the German Study Group on Gastrointestinal Lymphoma (DSGL). Strahlenther Onkol. 2019 Jun;195(6):544-557. English. doi: 10.1007/s00066-019-01446-5. Epub 2019 Mar 11. PMID: 30859254.)

Response : This comment and the corresponding reference have been added (page 11).

In parallel to the radiation dose reduction, the attempts to reduce the target volume have been undertaken, with the aim to reduce the toxicity and to improve tolerability of a modern radiotherapy in GML (Reinartz et al, 2019)

Recommendation:

Overall an interesting review underlining the importance of accurate diagnostics, adequate treatment strategies and long-term surveillance in GML patients.

Major changes are strongly necessary. I recommend revision of the document with explicit regard to the above mentioned suggestions and re-submission

Response: Thank you for the comments. Indeed, we recognize the importance of radiotherapy in the management of the patients with GML However, since we wished to focus our review on gastroenterological aspects, and since other excellent reviews centered on radiotherapy have been already published by expert groups, we did not develop more this aspect. We thank the Reviewer for the comments that allowed us to enrich our review by developing this aspect.

Round 2

Reviewer 1 Report

The authors revised the manuscript well in accordance with the reviewer’s suggestions, but some points still remain to be modified.

Minor comments

1.I could not recognize the findings of “tree-like-appearance” in the images of Fig 2c and 2d. If the authors could detect such findings, the characterized area should be indicated by arrows. Furthermore, the authors describe "the same images in BLI and high magnification (zoom)" in figure legend. However, the images of Fig 2c and 2d seem to be same magnification based on the scale of upper right. The authors need to address these concerns.

2.In Fig1 and Fig2, the authors should black out the words outside the image.

Author Response

Responses to the Reviewers (second round)

Reviewer 1

The authors revised the manuscript well in accordance with the reviewer’s suggestions, but some points still remain to be modified.

Minor comments:

1.I could not recognize the findings of “tree-like-appearance” in the images of Fig 2c and 2d. If the authors could detect such findings, the characterized area should be indicated by arrows. Furthermore, the authors describe "the same images in BLI and high magnification (zoom)" in figure legend. However, the images of Fig 2c and 2d seem to be same magnification based on the scale of upper right. The authors need to address these concerns.

Response: The arrows indicated the “tree-like appearance” have been added on image b), and the  legend has been modified accordingly:

Figure 2 : Endoscopic aspects of gastric MALT lymphoma by virtual chromoendoscopy. (a) While light endocopy : only very slight lesins are visible (indicated by arrows) ; (b) Endoscopic image in virtual chromoendoscopy by BLI (Blue Light Imaging) and magnification : the lesions are much better visible, with clearly abnormal vascular pattern showing a « tree-line appearance »  (arrows) with abnormal vessels organized as a trunk of the tree with its branches, and linear erosions ; (c) and (d) : the same images in BLI and magnification, focused on linear erosions.

2.In Fig1 and Fig2, the authors should black out the words outside the image.

Response: The words have been blacked out.

Reviewer 2 Report

The authors have carefully and extensively edited the submitted comments.

I recommend to accept in present form.

Author Response

Responses to the Reviewers (second round)

 Reviewer 2

The authors have carefully and extensively edited the submitted comments.

I recommend to accept in present form.

Response : Thank you
